# Mental Health of Healthcare Workers during COVID-19 Pandemic in Taiwan: The First Wave Outbreak Occurred 1 Year Later Than in Other Countries

**DOI:** 10.3390/ijerph20032623

**Published:** 2023-02-01

**Authors:** Lee-Ling Huang, Hsin-Cheng Chung, Li-Ling Huang, Shu-Yuan Cheng, Chuan-Hsiung Lin, Te-Feng Yeh

**Affiliations:** 1Department of Nursing, Cheng Ching General Hospital, Taichung 400620, Taiwan; 2Administration Center, Cheng Ching General Hospital, Taichung 400620, Taiwan; 3Department of Healthcare Administration, Central Taiwan University of Science and Technology, Taichung 406053, Taiwan

**Keywords:** COVID-19, coping behaviors, psychological, traumatic stress, healthcare workers

## Abstract

We probed the psychological influence exerted on traumatic stress endured by healthcare workers (HCWs) and the coping behaviors adopted during the first wave of COVID-19 in Taiwan, which occurred one year later than in other countries. Clinical HCWs from two branches of a hospital network in Taichung, Taiwan, were recruited for this cross-sectional study. The participants were administered a questionnaire on sociodemographic and work-related characteristics, perceived influence exerted by COVID-19, coping behaviors in relation to COVID-19, and Impact of Event Scale-Revised scores. We obtained 769 valid questionnaires. A chi-square test, generalized linear modeling, and multivariate stepwise regression analyses were performed. Although the first wave of COVID-19 occurred one year later in Taiwan than in other countries, the traumatic stress experienced by Taiwanese HCWs was noted to be comparable to that of those in other countries. Factors for increased traumatic stress included caring for more patients with COVID-19, fair or poor self-rated mental health, higher perceived influence of COVID-19, vulnerable household income, and more negative coping behaviors. Positive coping behaviors such as exposure reduction and protection measures decreased traumatic stress. Accordingly, managers should strengthen protective measures, enhance COVID-19-related training, and provide psychological support and counseling for high-risk employees.

## 1. Introduction

Shortly after December 2019, the time at which the first SARS-CoV-2 case was reported, this virus rapidly spread globally, and frequent mutations have led to multiple waves of COVID-19 outbreaks and recovery. By April 2021, the number of people infected with the virus exceeded 150 million, with more than 3.2 million deaths [1]. The pandemic had a different course in Taiwan. Due to being close to China, Taiwan was predicted to become the second most vulnerable region to the COVID-19 pandemic [2]. However, by April 2021, Taiwan had only 1163 infected cases and 15 deaths [3]. Due to Taiwan’s experience in dealing with the SARS epidemic, the Taiwanese government quickly established a set of proactive measures against SARS-CoV-2. On 20 January 2020, when the pandemic broke out in Wuhan, the Central Epidemic Command Center (CECC) was activated even before the first cases of COVID-19 occurred in Taiwan. The CECC’s objectives were to unify the deployment of pandemic response resources and to oversee the implementation of multiple restrictive pandemic response measures, with the emphasis placed on access control, contact tracing, testing, and quarantine and isolation [2,4]. The response measures initially implemented by the CECC and relevant authorities entailed bolstering border controls, applying cloud technology to record personal travel histories for clinical use, collecting and distributing personal protective equipment for HCWs, and restricting HCW international travel. In addition, each hospital or medical institution was mandated to stockpile adequate quantities of personal protective equipment and to seal off most of their entrances in order to facilitate the task of taking every hospital visitor’s temperature as well as their TOCC (travel history, occupation, contact history, and cluster). Furthermore, every hospital’s staff and visitors were mandated to wear masks and to prevent nosocomial infections, hospitals and institutions were mandated to establish outdoor screening stations [2]. In 2020, Taiwan reported only 823 confirmed cases and nine deaths [4].

Taiwan’s pandemic prevention measures in 2020 were highly successful; the CECC gradually started easing quarantine policies, allowing foreign personnel and goods to enter the country. However, Taiwan’s community outbreak started in early May 2021, with the source case involving an airline pilot. This pilot reportedly failed to comply with the regulations and was diagnosed as having COVID-19. On 19 May 2021, the CECC raised the pandemic alert of the entire country to Level 3. The measures implemented included the suspension of all schools and the prohibition of gatherings in public places. Although no actual lockdown was announced, most people spontaneously avoided going out. To avoid overloading the testing and care capacity of hospitals and the medical system, the CECC urgently expanded its capacity to set up rapid testing stations, quarantine hotels, and isolation wards to respond, improve detection and care capacity, and address the growing community infection rate. At the peak of the pandemic, the maximum numbers of confirmed cases and deaths in a single day were 576 and 39, respectively. The pandemic situation subsided within two months, with the number of new infections dropping to between 10 and 20 per day. On 27 July 2021, because of the gradual easing of the pandemic, the pandemic alert was lowered to Level 2 [5,6]. From May to July 2021—the period of the first wave pandemic’s peak—14,574 new cases and 829 deaths occurred in Taiwan [3].

The COVID-19 pandemic has gravely affected healthcare workers’ (HCWs’) mental health, manifesting in a wide range of effects from distress to multiple mental health symptoms. A meta-analysis found that during the COVID-19 pandemic, 21.5%, 21.7%, and 22.1% of HCWs experienced posttraumatic stress disorder (PTSD), moderate depression, and anxiety, respectively; these figures were much higher than those in the general population [7]. Saragih et al. [8] indicated that during the same pandemic, the pooled prevalence rates derived for distress, depression, PTSD, and anxiety among HCWs worldwide were 37%, 37%, 49%, and 40%, respectively. These values were higher than those in the aforementioned meta-analysis, suggesting that the situation had worsened with the expansion of the pandemic. Although Taiwan faced its first wave one year later than other countries, as the number of confirmed and suspected cases of COVID-19 continued to rise, HCWs in Taiwan were facing increased work stress, which consequently exerted effects on their mental health. In the present study, we evaluated the psychological influence exerted on traumatic stress endured by HCWs and the coping behaviors adopted during the first wave of COVID-19 in Taiwan.

## 2. Materials and Methods

### 2.1. Study Setting and Participants

Cheng Ching General Hospital, established in March 1932, is the oldest private hospital in Taichung, Taiwan. In recent years, the Cheng Ching medical network has expanded to include two regional teaching hospitals, six district hospitals, one nursing home, and two group practice clinics. Among the two regional teaching hospitals, the Ping Deng Branch, the first to be established (in 1932), is a medium-sized community hospital with 350 beds, whereas the Chung Kang Branch, established in 1996, is a tertiary care hospital with 700 beds and 13 medical centers. Both hospitals were government-designated institutions for the care of patients with COVID-19. The Chung Kang Branch also takes care of patients transferred from northern Taiwan.

In this cross-sectional study, we administered a survey to all clinical HCWs employed in these two hospitals between 5 and 30 June 2021. Of the 263 and 593 HCWs in the two hospitals, 224 and 545, respectively, responded to the questionnaire (response rates: 85.2% and 91.9%, respectively).

### 2.2. Ethical Approval

This study complied with the regulations of Taiwan’s Human Subjects Research Act. Approval for our executed study was granted by Cheng Ching General Hospital’s Institutional Review Board (No. HP210014). The entirety of the study population recruited for participation in the survey granted written informed consent prior to their enrollment. The participants were informed that they could stop participating in the survey—which was anonymous, with the entirety of the collected information being kept confidential—whenever they pleased.

### 2.3. Measures

The study measures comprised participants’ coping behaviors in relation to COVID-19, the perceived influence exerted by COVID-19, sociodemographic as well as work-related characteristics, support environment, and Impact of Event Scale-Revised (IES-R) scores. These measures had already been adopted by another hospital survey executed in 2020 [9].

#### 2.3.1. Sociodemographic as Well as Work-Related Characteristics

For our executed study, we assessed the following sociodemographic characteristics: age; having dependent children; education level; sex; living with a spouse or partner, parents, in-laws, minor children, or adult children; having friends and relatives in quarantine; living alone; household income vulnerability; self-rated mental health; having a spouse or partner; and self-rated physical health. The work-related characteristics evaluated in this study were the following: branch, clinical experience during the severe acute respiratory syndrome (SARS) outbreak, participation in COVID-19-related training, years of clinical experience, contact with or caring for confirmed or suspected COVID-19 cases, and supervisor position, and, finally, occupation (physician or nurse).

#### 2.3.2. Support Environment

For our executed study, we categorized the support environment measure into the following: government policy support, hospital support, and family and colleague support. We instructed the participants to rate all items related to this measure on a 5-point Likert scale whose endpoints spanned from 0 (“never”) to 4 (“always”). We derived the overall Cronbach α value to be 0.866, with the Cronbach α values derived for the three domains being 0.703–0.863.

#### 2.3.3. Perceived Influence Exerted by COVID-19

By employing a 17-item scale that can be used to assess social stressors and psychological stressors, the present study evaluated the recruited participants’ perceptions of the influence exerted by COVID-19. We instructed the participants to rate all relevant items on a 5-point Likert scale whose endpoints spanned from 0 (“never”) to 4 (“always”). For this scale, higher scores signify a stronger perceived influence exerted by COVID-19. Furthermore, a prior study derived the following four factors after subjecting the aforementioned scale’s items to exploratory factor analysis: social isolation, less frequent social activities, worry about the pandemic being uncontrollable, and increased work stress. We derived the overall Cronbach α value to be 0.899, with the values derived for the four domains being 0.552–0.914.

#### 2.3.4. Coping Behaviors in Relation to COVID-19

We used a self-developed 20-item scale to measure coping behaviors in relation to COVID-19 among the HCWs. We instructed the participants to rate all of the items of the aforementioned scale on a 5-point Likert scale whose endpoints spanned from 0 (“never”) to 4 (“always”). Furthermore, a prior study derived the following four factors after subjecting the aforementioned scale’s items to exploratory factor analysis: negative avoidance, positive mindfulness, exposure reduction, and protection measures. We derived the overall Cronbach α value to be 0.811, with the values derived for the four domains being 0.615–0.864.

#### 2.3.5. IES-R Scores

Our executed study also employed the 22-item IES-R—which is a validated self-report scale for assessing PTSD, including disturbing memories, disorders related to trauma, and persistent negative emotions—for assessment [10]. We instructed the participants to rate all of the items of the aforementioned scale on a 5-point Likert scale whose endpoints ranged from 0 (“not at all”) to 4 (“extremely”). In addition to obtaining a total score (0–88 points; overall Cronbach α = 0.957), we also calculated scores for the IES-R subscales: intrusion (eight items, Cronbach α = 0.909), avoidance (eight items, Cronbach α = 0.894), and hyperarousal (six items, Cronbach α = 0.894).

We used the total IES-R score to identify PTSD risk. A total score of ≥33 has been proposed to provide the most accurate PTSD diagnosis [11]; in addition, a threshold score of 24 has been determined to be an indicator of a stress response that is clinically significant and can thus be used to establish PTSD as a clinical concern [12,13,14]. In this study, we considered both thresholds.

### 2.4. Statistical Analysis

We herein present data pertaining to IES-R scores (≥24 and ≥33) and sociodemographic as well as work-related characteristics as numbers and percentages. We also herein present data pertaining to coping behaviors in relation to COVID-19, the perceived influence exerted by COVID-19, support environment, and overall and subscale IES-R scores as means ± standard deviations. By executing a chi-square test, we compared the two hospital branches in terms of sociodemographic as well as work-related characteristics. A generalized linear model was used, after adjustment for sociodemographic and work-related characteristics, to determine differences in coping behaviors in relation to COVID-19, the perceived influence exerted by COVID-19, the support environment, and the IES-R score between the two branches. We executed multivariate stepwise regression and logistic regression to assess the associations of personal characteristics with IES-R scores, coping behaviors in relation to COVID-19, the perceived influence exerted by COVID-19, and the support environment. In addition, we executed the multivariate regression after adjustment for personal characteristics to probe the associations of IES-R scores with coping behaviors in relation to COVID-19, the perceived influence exerted by COVID-19, and the support environment.

## 3. Results

### 3.1. Findings for Sociodemographic as Well as Work-Related Characteristics

The participants’ mean age was 36.1 ± 10.7 years (42.4% of them were <29 years old), and 87.8% of them were women. Moreover, of the respondents, 76.0% had an undergraduate or higher degree. Regarding family characteristics, 35.6% of our participants had dependent children. Approximately half of the participants (48.4%) lived with their spouse or partner; in addition, some lived alone (18.6%), with minor or adult children (27.8% or 8.6%, respectively), or with their parents or in-laws (30.8%). Of all the responding HCWs, 63.1% and 67.0% were also noted in this study to have rated their physical and mental health in the questionnaire as “good” and “very good,” respectively. In addition, among the responding HCWs, 7.9% reported having a relative or friend in quarantine, and 35.5% reported the pandemic to have affected their household income (Table 1). HCWs at the Chung Kang Branch were younger, less likely to live with a spouse or partner or with adult children, and more likely to live alone.

Regarding work-related characteristics, the mean clinical experience of the participants was 13.2 ± 10.6 years. Specifically, among the responding HCWs, 55.4% reported receiving COVID-19-related training; 34.1% reported having been in contact with individuals with confirmed or suspected COVID-19 infection; 32.5% reported having 5–14 years of clinical experience (the most common number); 26.5% reported being engaged in clinical work during the SARS outbreak; and 10.1% and 94.3% reported being supervisors and nurses, respectively (Table 1). HCWs at the Chung Kang Branch had less clinical experience, were less likely to have had clinical experience during the SARS outbreak, and were more likely to have contact with individuals with confirmed or suspected COVID-19 infection.

### 3.2. Comparison of HCWs at Two Branches in Terms of Perceived Influence Exerted by COVID-19, Coping Behaviors in Relation to COVID-19, Support Environment, and IES-R Scores

Regarding the support environment, high support was obtained from family and colleagues (2.55). This was also the only domain with a mean score of >2; the mean scores derived for hospital support and government policy support were 1.86 and 1.74, respectively. After adjustment for sociodemographic and work-related characteristics, we noted that the scores derived for hospital support and government policy support for the HCWs at the Chung Kang Branch were significantly lower than those for the HCWs at the Ping Deng Branch (Table 2).

Concerning our responding HCWs’ perceived influence exerted by COVID-19, we determined that less frequent social activities had the highest mean score (3.23) and that social isolation had the lowest mean score (2.28). We noted the mean scores for worry about the pandemic being uncontrollable and for increased work stress to be 2.47 and 2.28, respectively. We executed adjustments for sociodemographic as well as work-related characteristics; subsequently, we determined that the HCWs at the Chung Kang Branch had significantly higher scores for social isolation and increased work stress than those at the Ping Deng Branch (Table 2).

The most common coping behavior was reduced exposure (3.21), and this was followed by protection measures (3.05) and positive mindfulness (2.27), with negative avoidance being the least common coping behavior (1.34). The coping behaviors were not significantly different between HCWs from the two branches.

Overall, we derived the mean IES-R score for our responding HCWs to be 22.12 ± 15.25. We noted the mean scores derived for the hyperarousal, avoidance, and intrusion to be 5.95 ± 4.41, 7.54 ± 5.76, and 8.63 ± 5.97, respectively. These four means were significantly higher in the HCWs at the Chung Kang Branch than those at the Ping Deng Branch. (Table 2) Moreover, regarding the IES-R scores among the responding HCWs, we derived a score of ≥24 in 295 HCWs (38.4%), signifying PTSD to be of clinical concern, and derived a score of ≥33 in 197 HCWs (25.6%), signifying the PTSD risk to be moderate or high risk. The percentages of HCWs with scores above the two cutoff points were significantly higher for the Chung Kang Branch than for the Ping Deng Branch (Figure 1).

### 3.3. Multivariate Analysis of Factors Associated with IES-R

In our executed multivariate stepwise regression, we observed the mean scores derived for the overall IES-R and for the three subscales to be significantly higher in HCWs with self-rated poor mental health, with affected household income, and working at the Chung Kang Branch. The mean scores for the intrusion subscale were significantly higher in the HCWs who had contact with patients with COVID-19. The scores for this subscale were significantly higher in nurses than in physicians. The mean scores for the avoidance subscale were significantly higher in those living with adult children and in nurses (Table 3).

According to our executed multivariate stepwise logistic regression, HCWs who were working at the Chung Kang Branch, reported living with adult children, had self-rated poor mental health, reported their household income was affected by the pandemic, and reported having contact with patients with COVID-19, were determined to be at a significantly higher risk of having an IES-R score of ≥24. Those who worked at the Chung Kang Branch, had self-rated poor mental health, and reported their household income was affected by the pandemic were also noted to be at a significantly higher risk of having an IES-R score of ≥33 (Table 4).

After adjustment for sociodemographic and work-related characteristics, multivariate regression analyses indicated that social isolation and worry about the pandemic being uncontrollable (under perceived influence exerted by COVID-19) and negative avoidance (under coping behaviors in relation to COVID-19) were significantly positively associated with overall IES-R and subscale scores, whereas exposure reduction (under coping behaviors in relation to COVID-19) was significantly negatively associated with these scores. Less frequent social activities (under perceived influence exerted by COVID-19) were significantly negatively associated with the overall score and scores in all subscales except for the hyperarousal subscale. Only the score in the intrusion subscale had no significant association with protection measures (under coping behaviors in relation to COVID-19). All other subscale scores had a significant negative association (Table 5). Moreover, worry about the pandemic being uncontrollable, social isolation, and negative avoidance were risk factors for protection measures, and exposure reduction (under coping behaviors in relation to COVID-19) were protective factors for an IES-R score ≥ 24. Furthermore, hospital support, the three domains of perceived influence exerted by COVID-19 (except less frequent social activities), and negative avoidance were significantly positively associated with less frequent social activities, protection measures, and exposure reduction were significantly negatively associated with IES-R score ≥ 33 (Table 6).

## 4. Discussion

The psychological impact exerted on HCWs by the pandemic may be due to the HCWs’ fear of infecting themselves and their family members or friends, their concerns about the pandemic being uncontrollable, their concerns about social isolation and stigma, unclear and inadequate protection policies and training, and inadequate preparedness. HCWs’ fear of lack of support from family, colleagues, hospitals, and the government may also contribute to the effects of the pandemic on their mental health [15,16,17]. We noted that less frequent social activities during the pandemic exerted the greatest effect on HCWs. In addition, the CECC announced the restriction of HCWs from traveling abroad on 23 February 2020, which meant that HCWs must cancel their travel plans and reduce contact with family members or relatives to prevent themselves from being infected and infecting others. We determined in this study that among the domains of coping behaviors in relation to COVID-19, exposure reduction had the highest mean score; protection measures also had a high mean score. HCWs avoided going out or using public transportation, maintained an appropriate social distance from others, followed protective measures and guidelines established by hospitals, and received education and training about COVID-19. We observed that negative avoidance had lower mean scores than other domains; relevant behaviors included emotional venting through smashing things, screaming, crying, cigarette smoking, drug use, or alcohol consumption. Consistent with the results of other studies, our data indicate that HCWs tended to adopt positive coping behaviors during COVID-19, which means that avoiding infection was their primary concern [15,16,17,18,19,20].

The overall mean IES-R score in our cohort of HCWs was 22.12 ± 15.25, and the mean scores for the intrusion, avoidance, and hyperarousal subscales were 8.63 ± 5.97, 7.54 ± 5.76, and 5.95 ± 4.41, respectively. Moreover, 38.4% and 25.6% of the HCWs obtained IES-R scores of ≥24 and ≥33, respectively. The risks of traumatic stress and PTSD in the HCWs do not differ considerably from those observed in other countries, although the peak of the pandemic occurred one year later [5,11,21,22,23,24,25,26,27,28,29,30,31,32,33,34,35]. This indicates that although HCWs in Taiwan had an extra year to prepare for the pandemic, when the pandemic became severe, the increase in the number of patients increased their care responsibilities, resulting in greater stress on the HCWs and negatively affecting their mental health.

In the present study, we noted risk factors for posttraumatic stress to be self-rated, fair or poor mental health, household income vulnerability, contact with patients with COVID-19, and being a nurse. The overall IES-R score and subscale scores were determined to be significantly higher in HCWs at the Chung Kang Branch than in those at the Ping Deng Branch. According to other studies employing the IES-R, HCWs with a history of mental health problems, self-reported history of a prior psychiatric condition, and preexisting psychological problems have higher overall IES-R and subscale scores [10,21,25,27,35,36]. Yang et al. [26] indicated that because lower-income groups are more likely to encounter economic difficulties, they are likely to experience higher psychological distress than higher-income groups. HCWs who had contact with patients with COVID-19 had higher IES-R scores than those who did not because direct exposure and increased work time and workload resulted in risks of infection and burnout [30,37,38,39]. Due to nurses having longer exposure to patients with COVID-19 than other HCWs, being a nurse is a significant risk factor for a high IES-R score [15,21,23,27,37,38,39,40].

According to our stepwise regression and logistic regression, worry about the pandemic being uncontrollable and social isolation (under perceived influence exerted by COVID-19), as well as negative avoidance (under coping behaviors in relation to COVID-19), were risk factors for a high IES-R score. Less frequent social activities (under perceived influence exerted by COVID-19) and exposure reduction and protection measures (under coping behaviors in relation to COVID-19) were protective factors against a high IES-R score. HCWs’ fear of infecting themselves and their family members and concerns about being stigmatized were determined to be risk factors for their poor mental health [26,27,38,40]. Positive coping behaviors can reduce psychological distress and include having adequate protection, education, training, and reduced social contact. Negative coping behaviors such as avoidance, self-blame, and indulgence have negative effects on psychological well-being [17,18,38,41,42,43].

Our executed study’s limitations warrant highlighting. First, this was a cross-sectional analysis of specific time points during the pandemic; hence, our results may not be generalizable to the pandemic as a whole. Future studies could examine the short- and long-term effects of different periods of the pandemic on the mental health of HCWs. Second, Taiwan’s first wave of the pandemic occurred a year later than that in other countries and may vary due to timing differences. Therefore, our data may not fully reflect Taiwanese HCWs’ mental health outcomes. Nevertheless, the findings of our study can help mental health professionals and policymakers worldwide understand HCWs’ psychological distress experienced during the pandemic.

## 5. Conclusions

We executed our research to probe the psychological influence exerted on traumatic stress endured by Taiwanese HCWs and the coping behaviors adopted during the first wave of the COVID-19 pandemic. We found that regarding perceived influence exerted by and coping behaviors in relation to COVID-19, infrequent social activities and reduced exposure had the highest mean scores in our survey. Although the peak of the pandemic in Taiwan occurred one year later than in other countries, the traumatic stress experienced by the HCWs was comparable to that experienced by those in other countries, with risk factors including household income vulnerability, fair or poor self-rated mental health, and contact with COVID-19 cases. Worrying about the pandemic being uncontrollable, social isolation, and negative avoidance resulted in significantly higher IES-R scores, whereas protection measures and exposure reduction appeared to reduce the HCWs’ traumatic stress. Healthcare sector leaders play a critical role in ensuring that HCWs remain mentally healthy and are provided the necessary support for coping with the impact of the pandemic. This support can be in the form of mental health support, psychological counseling, enhanced COVID-19-related training, strengthened protective measures, and measures to alleviate HCWs’ concerns about infecting their family members.

## Figures and Tables

**Figure 1 ijerph-20-02623-f001:**
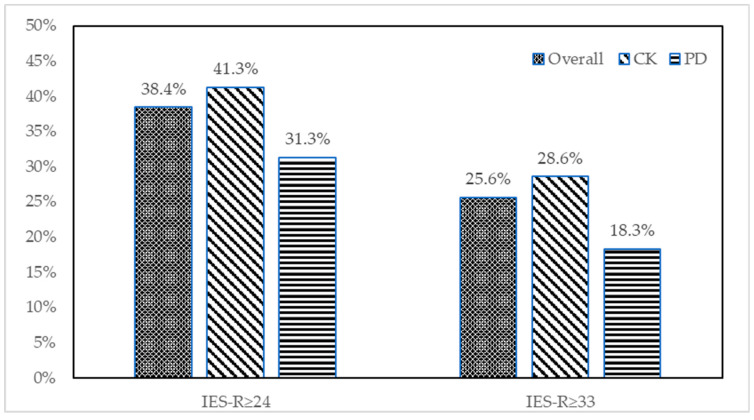
Comparison of IES-R scores between HCWs of the two hospitals.

**Table 1 ijerph-20-02623-t001:** Participants’ sociodemographic as well as work-related characteristics.

Personal Characteristic	Overall		CK		PD		χ^2^	*p*
	*n*	%	*n*	%	*n*	%		
**Total**	769		545		224			
**Sociodemographic characteristics**								
Women	675	87.8	473	86.8	202	90.2	1.700	0.192
Age (years)							14.302	0.003
<29	279	36.3	211	38.7	68	30.4		
30–39	197	25.6	147	27.0	50	22.3		
40–49	202	26.3	135	24.8	67	29.9		
>50	91	11.8	52	9.5	39	17.4		
Having dependent children	274	35.6	185	33.9	89	39.7	2.318	0.128
Living alone	143	18.6	114	20.9	29	12.9	6.663	0.010
Living with								
Spouse or partner	372	48.4	258	47.3	114	50.9	0.803	0.370
Parents or in-laws	237	30.8	169	31.0	68	30.4	0.032	0.859
Minor children	214	27.8	146	26.8	68	30.4	1.006	0.316
Adult children	66	8.6	33	6.1	33	14.7	15.234	0.000
Education level							1.429	0.490
College and below	185	24.1	125	22.9	60	26.8		
Undergraduate	541	70.4	388	71.2	153	68.3		
Graduate	43	5.6	32	5.9	11	4.9		
Excellent self-rated physical health	485	63.1	340	62.4	145	64.7	0.375	0.540
Excellent self-rated mental health	515	67.0	361	66.2	154	68.8	0.453	0.501
Quarantined relatives and friends	61	7.9	45	8.3	16	7.1	0.270	0.603
Household income vulnerability	273	35.5	202	37.1	71	31.7	1.998	0.158
**Work-related characteristics**								
Occupation							0.164	0.686
Physician	44	5.7	30	5.5	14	6.3		
Nurse	725	94.3	515	94.5	210	93.8		
Supervisor position	78	10.1	57	10.5	21	9.4	0.205	0.651
Years of clinical experience							9.958	0.019
<5	212	27.6	156	28.6	56	25.0		
5–14	250	32.5	190	34.9	60	26.8		
15–24	179	23.3	119	21.8	60	26.8		
>25	128	16.6	80	14.7	48	21.4		
Clinical experience during SARS	204	26.5	126	23.1	78	34.8	11.154	0.001
Contact with patients with COVID-19	262	34.1	204	37.4	58	25.9	9.409	0.002
Participation in COVID-19-related training	426	55.4	292	53.6	134	59.8	2.504	0.114

CK: Chung Kang Branch; PD: Ping Deng Branch.

**Table 2 ijerph-20-02623-t002:** Comparison of perceived influence exerted by COVID-19, coping behaviors in relation to COVID-19, support environment, and IES-R scores.

	Cronbach α	Overall		CK		PD		F	*p*
		Mean	SD	Mean	SE	Mean	SE		
**Support environment**	**0.866**								
Hospital support	0.863	1.86	0.72	1.82	0.09	1.99	0.10	8.060	0.005
Family and colleagues’ support	0.739	2.55	0.79	2.61	0.10	2.71	0.11	2.197	0.139
Government policy support	0.703	1.74	0.93	1.63	0.12	1.77	0.14	3.194	0.074
**Perceived influence exerted by COVID-19**	**0.899**								
Increased work stress	0.804	2.28	0.77	2.42	0.10	2.28	0.11	5.429	0.020
Worry about the pandemic being uncontrollable	0.914	2.47	0.85	2.39	0.11	2.35	0.12	0.477	0.490
Less frequent social activities	0.552	3.23	0.81	3.13	0.11	3.12	0.12	0.034	0.855
Social isolation	0.577	1.61	0.82	1.56	0.10	1.42	0.11	4.568	0.033
**Coping behaviors in relation to COVID-19**	**0.811**								
Protection measures	0.864	3.05	0.72	3.03	0.09	3.13	0.10	2.983	0.085
Exposure reduction	0.852	3.21	0.81	3.19	0.10	3.30	0.12	3.192	0.074
Positive mindfulness	0.838	2.27	0.71	2.18	0.09	2.24	0.10	0.930	0.335
Negative avoidance	0.615	1.34	0.69	1.39	0.09	1.28	0.10	3.783	0.052
**IES-R**	**0.957**	22.12	15.25	22.78	1.94	19.03	2.14	9.781	0.002
Intrusion	0.909	8.63	5.97	8.87	0.76	7.58	0.84	7.533	0.006
Avoidance	0.894	7.54	5.76	7.72	0.74	6.17	0.82	11.396	0.001
Hyperarousal	0.894	5.95	4.41	6.19	0.56	5.28	0.62	7.008	0.008

CK: Chung Kang Branch; PD: Ping Deng Branch; The model was adjusted for the participants’ sociodemographic and work-related characteristics.

**Table 3 ijerph-20-02623-t003:** Stepwise regression analysis results for IES-R-related factors.

	B(95% CI)	R^2^	Adj. R^2^	∆R2	
**IES-R**					
Excellent self-rated mental health	−6.466 (−8.693, −4.239)	0.048	0.047	0.048	***
Household income vulnerability	3.99 (1.799, 6.181)	0.065	0.062	0.017	***
Chung Kang Branch	3.928 (1.64, 6.216)	0.078	0.075	0.014	**
**IES-R-Intrusion**					
Excellent self-rated mental health	−2.177 (−3.048, −1.307)	0.043	0.042	0.043	***
Household income vulnerability	2.156 (1.296, 3.016)	0.070	0.067	0.027	***
Chung Kang Branch	1.282 (0.386, 2.177)	0.082	0.078	0.012	**
Contact with patients with COVID-19	1.186 (0.323, 2.048)	0.090	0.085	0.008	*
Physician	−1.761 (−3.518, −0.003)	0.094	0.088	0.005	*
**IES-R-Avoidance**					
Excellent self-rated mental health	−2.022 (−2.87, −1.174)	0.031	0.029	0.031	***
Chung Kang Branch	1.745 (0.866, 2.623)	0.046	0.044	0.016	***
Living with adult children	2.523 (1.052, 3.994)	0.058	0.054	0.012	**
Physician	−2.1 (−3.862, −0.337)	0.063	0.059	0.005	*
Household income vulnerability	0.894 (0.053, 1.735)	0.069	0.063	0.005	*
**IES-R-Hyperarousal**					
Excellent self-rated mental health	−2.158 (−2.801, −1.516)	0.062	0.060	0.062	***
Household income vulnerability	1.091 (0.459, 1.723)	0.076	0.074	0.015	***
Chung Kang Branch	0.949 (0.289, 1.609)	0.086	0.082	0.010	**

* *p* < 0.05; ** *p* < 0.01; *** *p* < 0.001.

**Table 4 ijerph-20-02623-t004:** Stepwise logistic regression analysis results for IES-R-related factors.

	B	SE	OR	95.% C. I.	*p*
**IES-R score ≥ 24**					
Chung Kang Branch	0.426	0.176	1.530	(1.083, 2.162)	0.016
Living with adult children	0.582	0.270	1.790	(1.055, 3.037)	0.031
Excellent self-rated mental health	−0.684	0.161	0.505	(0.368, 0.692)	0.000
Household income vulnerability	0.522	0.159	1.685	(1.233, 2.302)	0.001
Contact with patients with COVID-19	0.336	0.161	1.400	(1.020, 1.920)	0.037
**IES-R score ≥ 33**					
Chung Kang Branch	0.549	0.201	1.732	(1.168, 2.570)	0.006
Excellent self-rated mental health	−0.815	0.174	0.443	(0.315, 0.622)	0.000
Household income vulnerability	0.553	0.173	1.738	(1.237, 2.442)	0.001

**Table 5 ijerph-20-02623-t005:** Multivariate regression analysis of the associations of IES-R scores with coping behaviors in relation to COVID-19, the perceived influence exerted by COVID-19, and the support environment.

	IESR		IES-R-Intrusion		IES-R-Avoidance		IES-R-Hyperarousal	
	B(95%CI)		B(95%CI)		B(95%CI)		B(95%CI)	
Intercept	8.122 (2.471, 13.772)		1.911 (−0.314, 4.136)		3.468 (1.249, 5.687)		2.684 (0.971, 4.398)	
**Support environment**								
Hospital support	0.673 (−0.751, 2.097)		0.25 (−0.314, 0.815)		0.276 (−0.284, 0.836)		0.180 (−0.252, 0.612)	
Family and colleagues’ support	0.896 (−0.487, 2.279)		0.331 (−0.214, 0.876)		0.063 (−0.479, 0.605)		0.412 (−0.007, 0.832)	
Government policy support	−0.428 (−1.467, 0.611)		−0.143 (−0.553, 0.268)		−0.076 (−0.485, 0.332)		−0.206 (−0.521, 0.109)	
**Perceived influence exerted by COVID-19**								
Increased work stress	0.687 (−0.662, 2.037)		0.118 (−0.413, 0.65)		0.405 (−0.125, 0.935)		0.081 (−0.329, 0.49)	
Worry about the pandemic being uncontrollable	4.150 (2.775, 5.525)	***	2.142 (1.601, 2.684)	***	0.985 (0.445, 1.525)	***	1.028 (0.611, 1.445)	***
Less frequent social activities	−1.490 (−2.769, −0.211)	*	−0.531 (−1.035, −0.028)	*	−0.628 (−1.129, −0.127)	*	−0.299 (−0.687, 0.089)	
Social isolation	3.926 (2.636, 5.216)	***	1.338 (0.828, 1.847)	***	1.629 (1.121, 2.137)	***	1.017 (0.626, 1.408)	***
**Coping behaviors in relation to COVID-19**								
Protection measures	−1.552 (−3.027, −0.076)	*	−0.531 (−1.116, 0.055)		−0.770 (−1.349, −0.191)	**	−0.278 (−0.725, 0.17)	
Exposure reduction	−2.687 (−3.891, −1.483)	***	−0.986 (−1.46, −0.511)	***	−1.059 (−1.533, −0.586)	***	−0.714 (−1.079, −0.349)	***
Positive mindfulness	0.131 (−1.358, 1.619)		0.014 (−0.573, 0.6)		0.399 (−0.186, 0.984)		−0.347 (−0.799, 0.104)	
Negative avoidance	8.033 (6.668, 9.398)	***	2.868 (2.33, 3.406)	***	2.900 (2.364, 3.435)	***	2.285 (1.871, 2.699)	***
R^2^	0.445		0.443		0.443		0.391	
Adj. R^2^	0.435		0.431		0.431		0.380	
∆R^2^	0.367	***	0.349	***	0.349	***	0.305	***
F	43.211	***	37.410	***	31.841	***	34.622	***

* *p* < 0.05; ** *p* < 0.01; *** *p* < 0.001; model adjustment executed for the HCW sociodemographic as well as work-related characteristics.

**Table 6 ijerph-20-02623-t006:** Multivariate logistic regression analysis of the associations of IES-R scores with coping behaviors in relation to COVID-19, the perceived influence exerted by COVID-19, and the support environment.

Variable	IES-R Score ≥ 24		IES-R Score ≥ 33	
	OR (95%CI)		OR (95%CI)	
**Support environment**				
Hospital	0.915 (0.663, 1.264)		1.585 (1.091, 2.305)	*
Family and colleague	1.135 (0.836, 1.540)		0.913 (0.634, 1.315)	
Government policy	0.913 (0.726, 1.149)		0.776 (0.586, 1.027)	
**Perceived influence exerted by COVID-19**				
Increased work stress	1.168 (0.863, 1.580)		1.539 (1.056, 2.241)	*
Worry about the pandemic being uncontrollable	1.939 (1.423, 2.641)	***	1.938 (1.304, 2.881)	**
Less frequent social activities	0.817 (0.612, 1.092)		0.65 (0.459, 0.920)	*
Social isolation	1.687 (1.275, 2.232)	***	2.31 (1.655, 3.224)	***
**Coping behaviors in relation to COVID-19**				
Protection measures	0.649 (0.475, 0.886)	**	0.522 (0.367, 0.743)	***
Exposure reduction	0.652 (0.507, 0.839)	***	0.512 (0.385, 0.679)	***
Positive mindfulness	1.022 (0.737, 1.417)		1.223 (0.823, 1.817)	
Negative avoidance	3.07 (2.243, 4.200)	***	3.176 (2.219, 4.547)	***

* *p* < 0.05; ** *p* < 0.01; *** *p* < 0.001; model adjustment executed for HCW sociodemographic as well as work-related characteristics.

## Data Availability

The dataset that supports the findings of this study is not openly available and will be available from the corresponding author upon reasonable request.

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
