# Peer review of "Mental Health of Healthcare Workers during COVID-19 Pandemic in Taiwan: The First Wave Outbreak Occurred 1 Year Later Than in Other Countries"

_ijerph, 2023, doi:10.3390/ijerph20032623_

Round 1

Reviewer 1 Report

Clear and well written article.

Important and current topic.

Easy and pleasant to read, addressing an important and impactful topic for everyone, especially for health professionals.

In methodological terms, I consider it to be objective, revealing explicitly the methodological steps outlined and followed, including the ethical issues and precepts that were considered.

The title, although clear, could be even more specific if it referred to the studied population – Physician and nurses. I take the opportunity to question the reason for the exclusion of operational assistants.

In terms of keywords, I found that 3 met the criteria of the Mesh descriptors, missing “first wave pandemic” missing; “coping behaviors” and “psychological impact”. Maybe look for other replacement terms.

In terms of variables, they try to identify them and, at least, focus on how the more specific ones were operationalized, which is relevant.

Question in point 2.3.1 – what the difference between is “living with a spouse or partner” and “having a spouse or partner”

In point 3.2 I see no reference to Table 2.

In lines 250 to 253 I have doubts between the text and what appears in the table, specifically regarding the negative and positive statistically significant relationship.

Limitations are assumed, although it is not feasible to generalize the results to health professionals in Taiwan, it is a well-structured and methodologically well-designed study that, with the results obtained, can be an asset for the prevention of suffering psychological and for the promotion of the mental health of health professionals.

About bibliographical references, they present a very wide set of 43 references, with the exception of one dating from 2003 and which is related to statistical issues, all the others present a date equal to or greater than 2020 and all adjusted to the subject under study. However, and within the scope of standardization, I suggest rectifying references 12, 17, 33 and 36.

Author Response

Clear and well written article.

Thanks for this comment.

Important and current topic.

Thanks for this comment.

Easy and pleasant to read, addressing an important and impactful topic for everyone, especially for health professionals.

Thanks for this comment.

In methodological terms, I consider it to be objective, revealing explicitly the methodological steps outlined and followed, including the ethical issues and precepts that were considered.

Thanks for this comment.

The title, although clear, could be even more specific if it referred to the studied population – Physician and nurses. I take the opportunity to question the reason for the exclusion of operational assistants.

Thanks for this comment. In the operation of Taiwanese hospitals, physicians and nurses are the largest number of HCWs. Considering the project budget and the convenience of data collection, we only collected clinical practitioners of these two professions.

In terms of keywords, I found that 3 met the criteria of the Mesh descriptors, missing “first wave pandemic” missing; “coping behaviors” and “psychological impact”. Maybe look for other replacement terms.

Thanks for this comment. After re-searching MeSH descriptors, we retained “coping behaviors” (synonyms of “Adaptation, Psychological”), modified “psychological impact” to “psychological”, and deleted “first wave pandemic” (synonyms of “COVID-19”). (Line 28-29)

In terms of variables, they try to identify them and, at least, focus on how the more specific ones were operationalized, which is relevant.

Thanks for this comment. Since the measures for this study had already been performed at another hospital in the previous year, for considering paper similarity, we cited that publication(reference 9) and simplify the description of this study. (Line 114-118)

Question in point 2.3.1 – what the difference between is “living with a spouse or partner” and “having a spouse or partner”

Thanks for this comment. HCWs who have a spouse or partner who does not live with them because of work or various other reasons. Therefore, when designing the personal characteristics variable, we regard the two as different variables, and we can find that both were not associated factors in the stepwise models.

In point 3.2 I see no reference to Table 2.

Thanks for this comment. We include (Table 2) in each appropriate paragraph in 3.2. (Line 221,233,242)

In lines 250 to 253 I have doubts between the text and what appears in the table, specifically regarding the negative and positive statistically significant relationship.

Thank you for your kind reminder, we will review and correct the content. Except less frequent social activities was the only domain of perceived influence exerted by COVID-19 that was negatively associated with score ≥ 33. (Line 285-287)

Limitations are assumed, although it is not feasible to generalize the results to health professionals in Taiwan, it is a well-structured and methodologically well-designed study that, with the results obtained, can be an asset for the prevention of suffering psychological and for the promotion of the mental health of health professionals.

Thanks for this comment. Because the period and case numbers of the pandemic were different in different regions of Taiwan, and the number of cases in northern Taiwan is much higher than that in central Taiwan. Although the hospitals in this study also cared for COVID-19 patients, there was no guarantee that the psychological health of HCWs can fully reflect the facts about HCWs throughout Taiwan. We were therefore conservative about the generality of our findings.

About bibliographical references, they present a very wide set of 43 references, with the exception of one dating from 2003 and which is related to statistical issues, all the others present a date equal to or greater than 2020 and all adjusted to the subject under study. However, and within the scope of standardization, I suggest rectifying references 12, 17, 33 and 36.

Thanks for this comment. We review and rectify them again.

Reviewer 2 Report

1. The significance of the study is not supported in the introduction- what makes Taiwan important to study and different from all the other studies of HCWs that already dealt with COVID?  That has to be documented. 

2. The response rate is VERY high for a survey of busy healthcare workers- were incentives given?  

3. The first paragraph of the discussion seems more background than discussion of findings.

Author Response

  1. The significance of the study is not supported in the introduction- what makes Taiwan important to study and different from all the other studies of HCWs that already dealt with COVID?  That has to be documented. 

Thanks for this comment. We have moved the first two paragraphs from the Discussion to the Introduction to reinforce the background and value of the study. (Line 37-71)

  1. The response rate is VERY high for a survey of busy healthcare workers- were incentives given?  

Thanks for this comment. Because this study was an industry-academia cooperation project of the institutions to which the authors affiliate, and several authors are supervisors of the study sites, they put a lot of effort into the process of informed consent and questionnaire collection. Based on previous experience with low response rates to online surveys, we used documented informed consent and questionnaires, which were explained to respondents by IRB-approved hospital-affiliated members, who responded in sealed envelopes. The information was then processed by non-hospital-affiliated members to ensure the confidentiality of the respondents. Compared with the corresponding author's response rate (93.7%) in a similar study conducted in another hospital in the previous year, it is not very high.(reference 9)

  1. The first paragraph of the discussion seems more background than discussion of findings.

Thanks for this comment. We have moved the first two paragraphs from the Discussion to the Introduction to reinforce the background and value of the study. (Line 37-71)

Reviewer 3 Report

Dear authors, thank you very much for allowing me to read and evaluate this work. This research is a very interesting one and relevant at present.

Here are some suggestions for improvement:

It needs to include some literature review background rather than directly going to the research methodology section.

The research gap is not evident.

The validity of the questionnaire has been checked only using Cronbach's alpha?

What is the justification to use the specific statistical analysis?

It would be better to add the recommendation to the Health care sector regarding the mental health of the workers.

Thank you again for allowing me to read this paper. I hope that my contributions will be helpful to you.

Author Response

Dear authors, thank you very much for allowing me to read and evaluate this work. This research is a very interesting one and relevant at present.

Thanks for this comment.

Here are some suggestions for improvement:

It needs to include some literature review background rather than directly going to the research methodology section.

Thanks for this comment. We have moved the first two paragraphs from the Discussion to the Introduction to reinforce the background and value of the study. (Line 37-71)

The research gap is not evident.

Thanks for this comment. Because the evolution of the pandemic was different from other countries, HCWs in Taiwan faced the real first wave of the pandemic a year later than most countries. The value of this study lied in whether the mental health of HCWs in Taiwan was less impacted after more than a year of preparation, and it was found that HCWs in Taiwan had similar levels of traumatic stress as other countries in the early stages of the 2020 pandemic. (Line 37-71)

The validity of the questionnaire has been checked only using Cronbach's alpha?

Thanks for this comment. Since the measures for this study had already been performed at another hospital in the previous year, construct validity had been described in that publication (reference 9). For considering paper similarity, we cited that publication and only adopted Cronbach's alpha in this study.

What is the justification to use the specific statistical analysis?

According to several literatures, the study variables were different in the personal characteristics of different HCWs, and the personal characteristics of HCWs in the two branches of this study were also different, so statistical adjustments were made to avoid the influence of these variables.

It would be better to add the recommendation to the Health care sector regarding the mental health of the workers.

The last three sentences in the conclusion were recommendations for the health-care sector. In order to point out more specifically, we change Health-care leaders to Health-care sector leaders. (Line 410)

Thank you again for allowing me to read this paper. I hope that my contributions will be helpful to you.

Thanks for this comment.

Reviewer 4 Report

After reading the manuscript (Mental Health of Health-care Workers During COVID-19 Pandemic in Taiwan: The First Wave Outbreak Occurred 1 Year 3 Later than in Other Countries) it seems to me that it is an excellent candidate for publication at this journal. Authors probed the psychological influence exerted on, traumatic stress endured by, and coping behaviors adopted by health-care workers (HCWs) during the first wave of COVID-19 in Taiwan, which occurred 1 year later than in other countries. Clinical HCWs from two branches of a hospital network in Taichung, Taiwan, were recruited for this cross-sectional study. The participants were administered a questionnaire on socio-demographic and work-related characteristics, perceived influence exerted by COVID-19, and coping behaviors in relation to COVID-19, and Impact of Event Scale-Revised scores. I believe that the following comment may help the authors to revise it accordingly:

1) The title is amazing. It’s a good piece of work. It will be great if authors start the

abstract with mentioning the main object of the study.

2) The “1. Introduction: section is comparatively short. One more paragraph regarding

the importance of the study, objective and summary of the study can be discussed. You may add the novelty of the study, and how it contributes to the current literature. How your study enhances our understanding about health-care workers (HCWs) mental health and psychological health during Covid-19 pandemic? 

3) “2. Materials and Methods” is OK. The response rate is high, the authors have done a great job. Explain the reasons behind this achievement. It can help the other researchers in this context. Did you provide financial incentive or free meals or … 

4) “3. Results” Section is OK. Authors did a great job here. They designed the figures

and tables nicely that is eye soothing.

5) “4. Discussion” section is perfect.

6) “5. Conclusions” is OK.

7) It will be great if authors add the “implications of the study” as a separate paragraph.

8) If “limitations and directions for future studies” were added, it would be great. Hope

authors will add it as a separate paragraph in the revised version.

9) Authors should follow the referencing styles of the study properly.

Author Response

After reading the manuscript (Mental Health of Health-care Workers During COVID-19 Pandemic in Taiwan: The First Wave Outbreak Occurred 1 Year 3 Later than in Other Countries) it seems to me that it is an excellent candidate for publication at this journal. Authors probed the psychological influence exerted on, traumatic stress endured by, and coping behaviors adopted by health-care workers (HCWs) during the first wave of COVID-19 in Taiwan, which occurred 1 year later than in other countries. Clinical HCWs from two branches of a hospital network in Taichung, Taiwan, were recruited for this cross-sectional study. The participants were administered a questionnaire on socio-demographic and work-related characteristics, perceived influence exerted by COVID-19, and coping behaviors in relation to COVID-19, and Impact of Event Scale-Revised scores. I believe that the following comment may help the authors to revise it accordingly:

1) The title is amazing. It’s a good piece of work. It will be great if authors start the abstract with mentioning the main object of the study.

Thanks for this comment.

2) The “1. Introduction: section is comparatively short. One more paragraph regarding the importance of the study, objective and summary of the study can be discussed. You may add the novelty of the study, and how it contributes to the current literature. How your study enhances our understanding about health-care workers (HCWs) mental health and psychological health during Covid-19 pandemic? 

Thanks for this comment. We have moved the first two paragraphs from the Discussion to the Introduction to reinforce the background and value of the study. (Line 37-71)

3) “2. Materials and Methods” is OK. The response rate is high, the authors have done a great job. Explain the reasons behind this achievement. It can help the other researchers in this context. Did you provide financial incentive or free meals or … 

Thanks for this comment. Because this study was an industry-academia cooperation project of the institutions to which the authors affiliate, and several authors are supervisors of the study sites, they put a lot of effort into the process of informed consent and questionnaire collection. Based on previous experience with low response rates to online surveys, we used documented informed consent and questionnaires, which were explained to respondents by IRB-approved hospital-affiliated members, who responded in sealed envelopes. The information was then processed by non-hospital-affiliated members to ensure the confidentiality of the respondents. Compared with the corresponding author's response rate (93.7%) in a similar study conducted in another hospital in the previous year, it is not very high. (reference 9)

4) “3. Results” Section is OK. Authors did a great job here. They designed the figures and tables nicely that is eye soothing.

Thanks for this comment.

5) “4. Discussion” section is perfect.

Thanks for this comment. As suggested by other reviewers, we have moved the first two paragraphs from the Discussion to the Introduction to reinforce the background and value of the study. (Line 37-71)

6) “5. Conclusions” is OK.

Thanks for this comment.

7) It will be great if authors add the “implications of the study” as a separate paragraph.

Thanks for this comment. When writing this manuscript, we decided to present the implications of the study in Conclusion because of the journal's writing format. (Line 390-391)

8) If “limitations and directions for future studies” were added, it would be great. Hope authors will add it as a separate paragraph in the revised version.

Thanks for this comment. The last paragraph in Discussion is limitations of study, considering the current pattern of the pandemic, we also make recommendations for future studies.

9) Authors should follow the referencing styles of the study properly.

Thanks for this comment. We review and rectify them again.